# Taylor-Mode Automatic Differentiation for Higher-Order Derivatives in JAX

**Jesse Bettencourt**
University of Toronto & Vector Institute
jessebett@cs.toronto.edu

**Matthew J. Johnson**
Google Brain

**David Duvenaud**
University of Toronto & Vector Institute

## Abstract

One way to achieve higher-order automatic differentiation (AD) is to implement first-order AD and apply it repeatedly. This nested approach works, but can result in combinatorial amounts of redundant work. This paper describes a more efficient method, already known but with a new presentation, and its implementation in JAX. We also study its application to neural ordinary differential equations, and in particular discuss some additional algorithmic improvements for higher-order AD of differential equations.

## 1 Introduction

Automatic differentiation (AD) for Machine Learning is primarily concerned with the evaluation of first order derivatives to facilitate gradient-based optimization. The frameworks we use are heavily optimized to support this. In some instances, though much rarer, we are interested in the second order derivative information, e.g. to compute natural gradients. Third and higher order derivatives are even less common and as such our frameworks do not support their efficient computation. However, higher order derivatives may be considerably valuable to future research for neural differential equations, so efficiently computing them is critical.

The naïve approach to higher order differentiation, supported by most AD frameworks, is to compose derivative operation until the desired order is achieved. However, this nested-derivative approach suffers from computational blowup that is exponential in the order of differentiation. This is unfortunate because more efficient methods are known in the AD literature, in particular those methods detailed in Chapter 13 of Griewank and Walther [1] and implemented in [2] as arithmetic on Taylor polynomials.

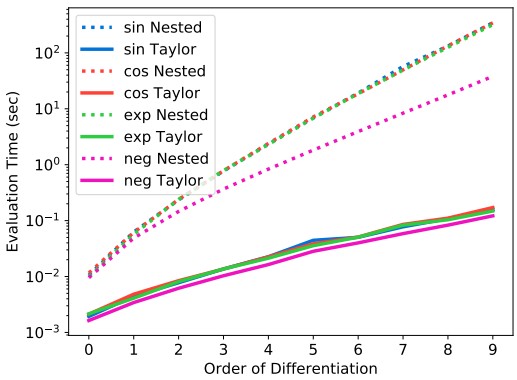

(a) Univariate primitives

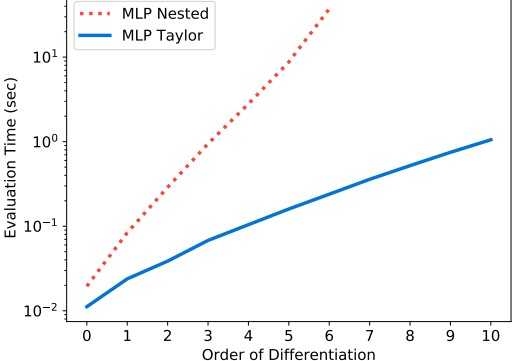

(b) Two-layer MLP with `exp` non-linearities

Figure 1: Scaling of higher-order AD with nested derivatives (dashed) vs Taylor-method (solid)

This Taylor-mode is a generalization of forward-mode AD to higher dimensional derivatives.

The exponential blowup can be seen in fig. 1 where dashed lines correspond to the naïve, nesting-derivative approach. Our implementation of Taylor-mode achieves much better scaling, as can be seen in fig. 1a where we show the higher-derivatives of some univariate primitives. The performance gains are even more dramatic in fig. 1b showing the scaling of higher-order derivatives for a 2-layer MLP with `exp` non-linearities.

In section 2 we give an overview of the problem of computing higher order derivatives. In section 3 we give an overview of our ongoing work to implement these methods in an AD framework for machine learning, namely `JAX`. In section 4 we demonstrate the relationship between these methods and solutions differential equations. In the appendix A we provide relevant mathematical background linking these mtehods to fundamental results from calculus.

## 2 Higher-Order Chain Rule

First-order automatic differentiation relies on solving a composition problem: for a function $f = g \circ h$, given a pair of arrays $(z_0, z_1)$ representing $(h(x), \partial h(x)[v])$, compute the pair $(f(x), \partial f(x)[v])$. We solve it using $f(x) = g(h(x)) = g(z_0)$ and $\partial f(x)[v] = \partial g(h(x)) [\partial h(x)[v]] = \partial g(z_0)[z_1]$.

A higher-order analogue of this composition problem is: given a tuple of arrays representing

$$(z_0, \ldots, z_K) = \left( h(x), \quad \partial h(x)[v], \quad \partial^2 h(x)[v, v], \quad \ldots, \quad \partial^K h(x)[v, \ldots, v] \right), \tag{1}$$

compute

$$\left( f(x), \quad \partial f(x)[v], \quad \partial^2 f(x)[v, v], \quad \ldots, \quad \partial^K f(x)[v, \ldots, v] \right). \tag{2}$$

We can solve this problem by developing a formula for each component $\partial^k f(x)[v, \ldots, v]$. The basic issue is that there are several ways in which to form $k$-th order perturbations of $f$, routed via the perturbations of $h$ we are given as input.

Take $k = 2$ for concreteness:

$$\partial^2 f(x)[v, v] = \partial g(z_0)[z_2] + \partial^2 g(z_0)[z_1, z_1]. \tag{3}$$

The first term represents how a 2nd-order perturbation in the value of $f$ can arise from a 2nd-order perturbation to the value of $h$ and the 1st-order sensitivity of $g$. Similarly the second term represents how 1st-order perturbations in $h$ can lead to a 2nd-order perturbation in $f$ via the 2nd-order sensitivity of $g$. For larger $k$, there are many more ways to combine the perturbations of $h$.

More generally, let $\mathrm{part}(k)$ denote the integer partitions of $k$, i.e. the set of multisets of positive integers that sum to $k$, each represented as a sorted tuple of integers. Then we have

$$\partial^k f(x)[v, \ldots, v] = \sum_{\sigma \in \mathrm{part}(k)} \mathrm{sym}(\sigma) \cdot \partial^{|\sigma|} g(z_0) \left[ z_{\sigma_1}, z_{\sigma_2}, \ldots z_{\sigma_{\mathrm{end}}} \right], \tag{4}$$

$$\mathrm{sym}(\sigma) := \frac{k!}{\sigma_1! \, \sigma_2! \, \cdots \, \sigma_{\mathrm{end}}!} \frac{1}{\prod_{i \in \mathrm{uniq}(\sigma)} \mathrm{mult}_\sigma(i)!} \tag{5}$$

where $|\sigma|$ denotes the length of the tuple $\sigma$, $\mathrm{uniq}(\sigma)$ denotes the set of unique elements of $\sigma$, and $\mathrm{mult}_\sigma(i)$ denotes the multiplicity of $i$ in $\sigma$.

An intuition for $\mathrm{sym}(\sigma)$ is that it counts the number of ways to form a perturbation of $f$ of order $k$ routed via perturbations of $h$ of orders $\sigma_1, \sigma_2, \ldots, \sigma_{\mathrm{end}}$ as a multinomial coefficient (the first term), then corrects for overcounting perturbations of $h$ of equal order (the second term). For further explanation of integer partitions, multiplicity, and intuition of $\mathrm{sym}$ refer to appendix B.

The problem of computing these higher derivatives, and the formula in (4) is known as the Faà di Bruno Formula. This can equivalently be expressed as the coefficients of a truncated Taylor polynomial approximation of $f$ at a point $x_0$.

$$f(x + v) \approx f(x) + \partial f(x)[v] + \frac{1}{2!} \partial^2 f(x)[v, v] + \cdots + \frac{1}{d!} \partial^d f(x)[v, \ldots, v]. \tag{6}$$

These connections are explained in background found in appendix A.

# 3 Implementation in JAX

Given the coefficients $x_0, \ldots, x_d$ as in the polynomial (12) we implement the function `jet` which computes the coefficients $y_0, \ldots, y_d$ as in the polynomial (13). The name refers to the jet operation described in eq. (16) of appendix A.3 with the interface

$$y_0, (y_1, \ldots, y_d) = \texttt{jet}\,(f, x_0, (x_1, \ldots, x_d)) \tag{7}$$

Supporting this user-facing API is a new JAX interpreter `JetTracer` which traces the forward evaluation and overloads primitive operations. As an aside, this is analogous to implementing forward-mode AD by overloading operations with dual number arithmetic. In fact, as noted in Rules 22 and 23 of Griewank and Walther [1], Taylor-mode with one higher derivative, corresponding to $f(x_0 + x_1 t)$, is identically forward-mode. However, `JetTracer` generalizes the usual forward-mode tracing to compute higher order jets.

This is achieved by overloading primitive operations to call an internal function `prop` that computes the higher order derivatives for each primitive operation. In particular, `prop` implements the Faà di Bruno algorithm (10). Internally it calls `sym` to compute the partition multiplicity for the integer combinatorial factors appearing in the higher derivatives.

Crucially, recall that Faà di Bruno expresses the total derivatives of the composition $f(g(x))$ in terms of the higher-order partial derivatives of $f$ and $g$. Further, the goal of this is to share computation of these partial derivatives across order of differentiation. To achieve this `prop` calls a generator which returns previously computed partial derivative functions. Again, this generalizes forward mode implementations which provide the first order derivative rule for each primitive.

**Example** The primitive `sin` is known to have a first order derivative `cos`. First-order forward-mode implementations stop providing derivative rules here. However, it is also known that the second order derivative is `-sin`, and third is `-cos`. So all higher derivatives of the primitive `sin` can be computed by overloading the primal evaluation. We can further exploit the shared evaluation at this level, i.e., all higher derivatives involve cycling through $\{\texttt{cos}, \texttt{-sin}, \texttt{-cos}, \texttt{sin}\}$, and even these only involve computing `sin` and `cos` once, and negating the result.

We implement these higher-order derivative rules for various primitives. While trigonometric primitives are an extreme example of sharing work across derivative order, many other primitives also can benefit from sharing some common sub-expressions.

There are two opportunities to share common work here. In defining the higher-order partial derivative rules for each primitive some evaluations can be shared. In computing the total derivative of function compositions partial derivatives of primitives should only be evaluated once for each order, then shared. The function `prop` is responsible for computing or accessing these partial derivatives at the necessary orders, computing their multiplicity with `sym`, and combing them according to the higher-order composition rule (4).

## 3.1 Comparing Taylor-Mode to Nested Derivatives

The nested approach to higher-order derivatives would involve taking the derivative of the primitive operations emitted by the first derivative evaluation. Consider again the above `sin` example, and its derivative `cos`. Nested-derivatives would evaluate the derivative of the emitted `cos` which would correctly return `-sin`. However, in addition to whatever overhead each additional derivative introduces, this higher derivative is not able to take advantage of the common evaluation from the lower order.

In fig. 1a we compare the computational scaling as we evaluate higher derivatives of various primitives. There it can be seen that the naïve nesting derivatives, by not sharing common expressions across orders, requires more computation for each subsequent derivative order. Whereas Taylor-mode enjoys much better scaling for the higher derivatives of the primitives.

This effect is considerably more dramatic in the case of higher derivatives of functions which compose primitives. In fig. 1b we show the scaling on a 2-layer MLP with `exp` non-linearities. Here it can be seen that nested derivatives scale much worse than the blowup from the individual primitives. Further, the nested derivatives were only able to be computed up to order 6 before memory limitations prevented higher differentiation. Taylor-mode was able to compute up to 10th order without experiencing memory limitations and in with significantly better scaling.

# 4 Differential Equations

Dynamical systems are defined directly in terms of their derivatives. Consider the initial value problem (IVP)

$$\frac{dx}{dt} = f(x(t)) \quad \text{where} \quad x(0) = x_0 \in \mathbb{R}^n$$

which we write in *autonomous* form for notational convenience, see appendix D.1

The exact solution to the IVP can be approximated by truncating the Taylor polynomial

$$x(t) = \sum_{i=0}^{d} x_i t^i + O(t^d) \in \mathbb{R}^n$$

$$= x_0 + \frac{dx}{dt}t + \frac{1}{2!}\frac{d^2 x}{dt^2}t^2 + \cdots + \frac{1}{d!}\frac{d^d x}{dt^d}t^d$$

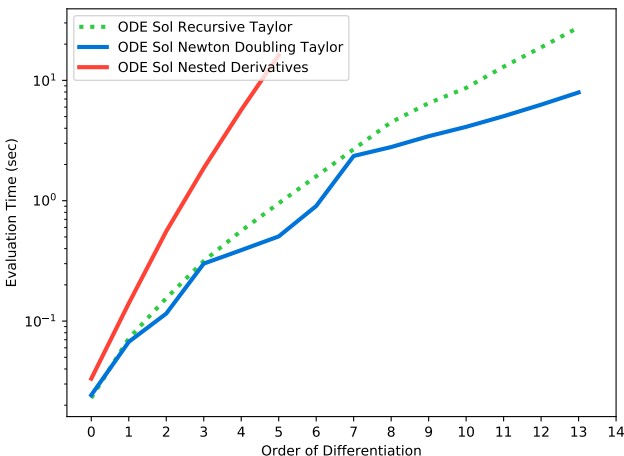

Figure 2: Methods for computing coefficients of an ODE Solution

In defining the IVP, we already specify the first coefficient to the Taylor polynomial, $x_1 = f(x(t))$. Further, we see that the second, and subsequent, coefficients must be higher derivatives with respect to $t$. Recall that these derivatives $x_{i=1...d} = \frac{d^i}{dt^i}f(x(t))$ are exactly those given by the coefficients of the polynomial $y(t) = f(x(t))$ (13).

This gives us the recursive relationship for computing the coefficients of the solution:

$$x_{i+1} = y_i \tag{8}$$

Recall that `jet`, by definition (16), gives us the coefficients for $y_i$ as a function of $f$ and the coefficients $x_{j \leq i}$. We can use `jet` and the relationship (8) to recursively compute the coefficients of the solution polynomial. Pseudocode for computing the coefficients of the ODE solution recursively with `jet` is given in appendix D.2 algorithm 2. However, a more efficient method is known which exploits linearity properties of higher derivatives.

## 4.1 Exploiting Linearity in Higher Coefficients of ODE Solution with Newton's Method

It is proven in Griewank and Walther [1] that the coefficient for an ODE solution $y_k = y_k(x_0, \ldots, x_k)$ is linear with respect to the upper half of its input coefficients, $x_j$ for $j > \frac{k}{2}$. Further, the lower half of its input coefficients, $x_0, \ldots, x_{j-1}$, determines the linear dependence as

$$x_{k+1} = y_k = y_k(x_0, \ldots, x_{j-1}, 0, \ldots, 0) + \frac{k!}{(j-1)!} \sum_{i=j}^{k} \frac{((j-1)-(k-i))!}{i!} A_{k-i} x_i \tag{9}$$

where $A_m = \frac{\partial y_{j-1}}{\partial x_{(j-1)-m}}$. See appendix D.3 for details.

In particular, notice that linear updates are given by a sum of Jacobian-vector products $\sum_{i=j}^{k} A_{k-i} x_i$. This means that we do not need to explicitly instantiate any Jacobians, since they are immediately contracted against vectors. Further, we also do not need to compute many Jacobian-vector products and then sum the results. As discussed in appendix D.4, we can compute this sum by a single Jacobian-vector product of $y_{j-1}$ with respect to all its inputs $x_0, \ldots, x_{j-1}$.

The following algorithm appears in Table 13.7 of [1], but we make it clear that Jacobians are represented implicitly through Jacobian-vector products in algorithm 1.

Figure 2 shows that exploiting higher-coefficients' linearity via Newton's Doubling Method allows for more efficient computation of coefficients except those which require full non-linear evaluation.

**Algorithm 1** ODE Solution by Doubling Coefficients through Newton's Method

---

```
# Have: x0,f
# Want: x1, ..., xd

for s in 0,1,...:
  j = 2**(s+1)-1

  # function computes nonlinear dependence on lower-half coefficients
  f_jet = lambda x0, ..., x{j-1} : jet(f, (x0,), ([x1, ..., x{j-1}, 0, ...,0])

  # linearize the function to get y_hat and function which computes jvps
  y_hat, jvp_jet = jax.linearize(f_jet,x0,...,x{j-1})

  for k in j-1 ... 2j-1:

    # update yhat with sum of jvps, computed in single jvp
    yk = y_hat[k] + k!/(j-1)! * jvp_jet(*[(j-1)-(k-i))!/i! xi for i in j...k])

    # recurrence relationship from ODE dynamics
    x{k+1} = yk

    if k+1==d:
      break

return x0, [x1, ..., xd]
```

---

## References

[1] Andreas Griewank and Andrea Walther. Evaluating derivatives. 2008.

[2] Andreas Griewank, David Juedes, and Jean Utke. Algorithm 755: Adol-c: a package for the automatic differentiation of algorithms written in c/c++. *ACM Transactions on Mathematical Software (TOMS)*, 22(2):131–167, 1996.

[3] Luis Benet and David Sanders. Taylorseries.jl: Taylor expansions in one and several variables in julia. *Journal of Open Source Software*, 4, 04 2019. doi: 10.21105/joss.01043.

[4] Winston C. Yang. Derivatives are essentially integer partitions. *Discrete Mathematics*, 222(1): 235 – 245, 2000. ISSN 0012-365X. doi: https://doi.org/10.1016/S0012-365X(99)00412-4. URL http://www.sciencedirect.com/science/article/pii/S0012365X99004124.

[5] Michael Hardy. Combinatorics of partial derivatives. *the electronic journal of combinatorics*, 13, Jan 2006. URL http://arxiv.org/abs/math/0601149v1. Electronic Journal of Combinatorics 13 (2006) #R1.

[6] Warren Johnson. The curious history of faa di bruno's formula. *The American Mathematical Monthly*, 109:217–234, 03 2002. doi: 10.1080/00029890.2002.11919857.

[7] Ernst Hairer, Syvert Norsett, and G. Wanner. *Solving Ordinary Differential Equations I: Nonstiff Problems*, volume 8. 01 1993. doi: 10.1007/978-3-540-78862-1.

# A  Background

## A.1  Faà di Bruno's Formula

If $f$ and $g$ are sufficiently smooth functions then the $n$th derivative of their composition $f(g(x))$ is well-known and given by the Faà di Bruno formula which generalizes the chain rule to higher derivatives:

$$\frac{d^n}{dx^n}f(g(x)) = \sum_{\sigma \in \pi_n} \frac{n!}{k_1! \cdots k_n!} f^{(k_1 + \cdots + k_n)}(g(x)) \left( \left(\frac{1}{1!}\frac{\partial g(x)}{\partial x}\right)^{k_1} \cdots \left(\frac{1}{n!}\frac{\partial^n g(x)}{\partial x^n}\right)^{k_n} \right)$$

$$= \sum_{\sigma \in \pi_n} \text{sym}(\sigma) f^{(\sum_i k_i)}(g(x)) \prod_{i:k_i \neq 0 \in \sigma} (\frac{d^i g(x)}{dx^i})^{k_i} \tag{10}$$

where $\pi_n$ is the set of all n-tuples $(k_1, \ldots, k_n)$ of non-negative integers such that $\sum_i i k_i = n$. We introduce the function $\text{sym}(\sigma)$ which computes the multiplicity associated with the partition $\sigma$. See appendix B for details on partition multiplicity and an example.

The Faà di Bruno algorithm (10) give the familiar expressions

$$
\begin{aligned}
\frac{d}{dx}f(g(x)) &= f'(g(x))\frac{dg(x)}{dx} \\
\frac{d^2}{dx^2}f(g(x)) &= f'(g(x))\frac{d^2 g(x)}{dx^2} + f''(g(x))\left(\frac{d^2 g(x)}{dx^2}\right)^2 \\
\frac{d^3}{dx^3}f(g(x)) &= f'(g(x))\frac{d^3 g(x)}{dx^3} + 3f''(g(x))\frac{dg(x)}{dx}\frac{d^2 g(x)}{dx^2} + f'''(g(x))\left(\frac{dg(x)}{dx}\right)^3
\end{aligned}
\tag{11}
$$

$$\vdots$$

Importantly, the expression for the $n$th order derivative of the composition $f(g(x))$ is given in terms of (mostly lower order) derivatives of the constituent functions $f$ and $g$. Since we are concerned with computing all derivatives of the composition up to $n$, this formula allows us to share the work of computing lower-order derivatives with all subsequent higher orders. For example, computing the derivative $\frac{d^3 f(g(x))}{dx^3}$ requires the value of $\frac{d^2 g(x)}{dx^2}$ which was already computed for the previous derivative $\frac{d^2 f(g(x))}{dx^2}$. The Faà di Bruno algorithm makes explicit how these intermediate quantities can be shared across the orders of differentiation.

## A.2  Taylor Polynomials

Truncated Taylor polynomials allow for natural representation and manipulation of higher order derivatives, and the relationship between their polynomial arithmetic and AD for higher derivatives is well-known [1].

Consider the polynomial

$$x(t) = x_0 + x_1 t + \frac{1}{2!}x_2 t^2 + \frac{1}{3!}x_3 t^3 + \cdots + \frac{1}{d!}x_d t^d \in \mathbb{R}^n \tag{12}$$

For a sufficiently smooth vector valued function $f : \mathbb{R}^n \to \mathbb{R}^m$, we are interested in the truncated Taylor polynomial given by the resulting expansion

$$y(t) \equiv y_0 + y_1 t + \frac{1}{2!}y_2 t^2 + \frac{1}{3!}y_3 t^3 + \cdots + \frac{1}{d!}y_d t^d \in \mathbb{R}^m \tag{13}$$

the coefficients $y_j$ of which are smooth functions of the $i \leq j$ coefficients $x_i$:

$$
\begin{aligned}
y_0 &= y_0(x_0) & &= f(x_0) \\
y_1 &= y_1(x_0, x_1) & &= f'(x_0)x_1 \\
y_2 &= y_2(x_0, x_1, x_2) & &= f'(x_0)x_2 + f''(x_0)x_1 x_1 \\
y_3 &= y_3(x_0, x_1, x_2, x_3) & &= f'(x_0)x_3 + 3f''(x_0)x_1 x_2 + f'''(x_0)x_1 x_1 x_1
\end{aligned}
\tag{14}
$$

$$\vdots$$

If we allow that $x(t)$ itself is a Taylor polynomial, with the suggestive notation that its coefficients capture higher derivatives of its dependence on the independent variable $t$, i.e. $x_i = \frac{d^i x(t)}{dt^i}$, then the meaning of the coefficients of $y(t)$ become clear:

$$
\begin{aligned}
y_0 &= f(x_0) \\
y_1 &= f'(x_0)\frac{dx}{dt} & &= \frac{d}{dt}f(x(t)) \\
y_2 &= f'(x_0)\frac{d^2 x}{dt^2} + f''(x_0)\left(\frac{dx}{dt}\right)^2 & &= \frac{d^2}{dt^2}f(x(t)) \\
y_3 &= f'(x_0)\frac{d^3 x}{dt^3} + 3f''(x_0)\frac{dx}{dt}\frac{d^2 x}{dt^2} + f'''(x_0)\left(\frac{dx}{dt}\right)^3 & &= \frac{d^3}{dt^3}f(x(t)) \\
&\ \ \vdots
\end{aligned}
\tag{15}
$$

That is, the coefficients $y_i$ are exactly the $i$th order derivative of the composition $f(x(t))$ with respect to $t$. Further, their intermediate expansions exactly correspond to the expressions for the higher order derivatives given by Faà di Bruno's Formula, for example compare (15) to (11). We refer to the coefficients of $y(t)$ as *derivative* coefficients. Refer to appendix C.1 for a comparison to how these are presented in [1], which equivalently incorporates the factorial terms but obfuscates their meaning as higher derivatives and their relationship with the Faà di Bruno Formula.

Our implementation makes extensive use of the relationship between eq. (15) and Faà di Brunno's Formula. Previous work on higher-order automatic differentiation using Taylor series instead rely directly on polynomial arithmetic of the truncated polynomials [1, 3]. While polynomial arithmetic will also give equivalent coefficients as eq. (15) it does not make explicit how computation should be shared across order of differentiation.

### A.3 Jets

The introduction of the Taylor polynomials in eqs. (12) and (13) are useful for relating this to mathematical foundations and to implementations which explicitly use polynomial arithmetic. However, while elegant, representing higher derivatives by polynomials introduces the independent variable, $t$, which is potentially a subtle confusion.

To clarify, we borrow the language of jets, an operation, $J_{x_0}^d$, on differentiable functions $f : X \to Y$ that produces the d-truncated Taylor polynomial of $f$ at every point $x_0$ in its domain. This is a useful operation because it allows us to consider jets as abstract polynomials in $x_0$, not as literal polynomials in the introduced independent variable $t$.

This view makes it clear that the functional dependency is on, $x_0$, where the polynomial is developed, not $t$, where it is evaluated. To relate jets to polynomials $x(t)$ (12) and $y(t)$ (13) we write:

$$
(J_{x_0}^d f)(x_1, \ldots, x_d) = y_0, \ldots, y_d
\tag{16}
$$

## B   More on Faà di Bruno's Formula

If $f$ and $g$ are sufficiently smooth functions then the $n$th derivative of their composition $f(g(x))$ is well-known and given by the Faà di Bruno formula which generalizes the chain rule to higher derivatives:

$$
\frac{\partial^n}{\partial x_1 \cdots \partial x_n}f(g(x)) = \sum_{\sigma \in \pi_{\{1,\ldots,n\}}} f^{(|\sigma|)}(g(x)) \prod_{b \in \sigma} \frac{\partial^{|b|}}{\prod_{j \in b}\partial x_j}g(x)
\tag{17}
$$

where $\pi_{\{1,\ldots,n\}}$ is the set of all partitions of the set $\{1, \ldots, n\}$.

The Faà di Bruno formula relates the $n$th derivative of a function composition with the combinatorial problem of how to combine the various lower-order partial derivatives. This is naturally described in terms of partitions of integer sets [4, 5] on the order of the desired derivative $n$, though there are other interpretations of the formula [6].

**Example** Consider the third derivative of the composition $f(g(x))$. Faà di Bruno gives that this will be related to the partitions of the set $\{1, 2, 3\}$, which is

$$
\pi_{\{1,\ldots,n\}} = \{ \\
\{\{1, 2, 3\}\}, \\
\{\{1\}, \{2, 3\}\}, \{\{2\}, \{1, 3\}\}, \{\{3\}, \{1, 2\}\}, \\
\{\{1\}, \{2\}, \{3\}\} \\
\}
$$

and the forumula 17 gives:

$$
\frac{\partial^3}{\partial x_1 \partial x_2 \partial x_3} f(g(x)) = f^{(1)}(g(x)) \frac{\partial^3 g(x)}{\partial x_1 \partial x_2 \partial x_3}
$$
$$
+ f^{(2)}(g(x))(\frac{\partial g(x)}{\partial x_1} \frac{\partial^2 g(x)}{\partial x_2 \partial x_3} + \frac{\partial g(x)}{\partial x_2} \frac{\partial^2 g(x)}{\partial x_1 \partial x_3} + \frac{\partial g(x)}{\partial x_3} \frac{\partial^2 g(x)}{\partial x_1 \partial x_2})
$$
$$
+ f^{(3)}(g(x)) \frac{\partial g(x)}{\partial x_1} \frac{\partial g(x)}{\partial x_2} \frac{\partial g(x)}{\partial x_3} \tag{18}
$$

However, it is often the case that the higher-order derivatives are taken with respect to the same indistinguishable variable, so $\frac{\partial^n}{\partial x_1 \cdots \partial x_n}$ becomes $\frac{d^n}{dx^n}$. In this case the Faà di Bruno formula 17 can be expressed in terms of partitions of the integer $n$ and the multiplicity of the parition elements:

$$
\frac{d^n}{dx^n} f(g(x)) = \sum_{\sigma \in \pi_n} \frac{n!}{k_1! \cdots k_n!} f^{(k_1 + \cdots + k_n)}(g(x)) \left( \left( \frac{1}{1!} \frac{\partial g(x)}{\partial x} \right)^{k_1} \cdots \left( \frac{1}{n!} \frac{\partial^n g(x)}{\partial x^n} \right)^{k_n} \right)
$$
$$
= \sum_{\sigma \in \pi_n} \frac{n!}{k_1! \cdots k_n! \cdot 1!^{k_1} \cdots n!^{k_n}} f^{(\sum_i k_i)}(g(x)) \left( \left( \frac{\partial g(x)}{\partial x} \right)^{k_1} \cdots \left( \frac{\partial^n g(x)}{\partial x^n} \right)^{k_n} \right)
$$
$$
= \sum_{\sigma \in \pi_n} \frac{n!}{k_1! \cdots k_n! \cdot 1!^{k_1} \cdots n!^{k_n}} f^{(\sum_i k_i)}(g(x)) \prod_{i : k_i \neq 0 \in \sigma} (\frac{d^i g(x)}{dx^i})^{k_i}
$$
$$
= \sum_{\sigma \in \pi_n} \mathrm{sym}(\sigma) f^{(\sum_i k_i)}(g(x)) \prod_{i : k_i \neq 0 \in \sigma} (\frac{d^i g(x)}{dx^i})^{k_i} \tag{19}
$$

Where $\pi_n$ is the set of all n-tuples $(k_1, \ldots, k_n)$ of non-negative integers such that $\sum_i i k_i = n$. We introduce the function $\mathrm{sym}(\sigma)$ which computes the multiplicity associated with the partition $\sigma$, that is, how many partitions of $\pi_{\{1,\ldots,n\}}$ that $\sigma$ corresponds to in $\pi_n$.

**Example** Consider again the third derivative of the composition $f(g(x))$. The integer partitions of $n = 3$ are

$$
\pi_3 = \{(0, 0, 1), (1, 1, 0), (3, 0, 0)\}
$$

The only partition in $\pi_3$ with non-trivial multiplicity is $(1, 1, 0)$ which corresponds to the partition $3 = 1 + 2 * 1 + 3 * 0$. $(1, 1, 0)$ identifies the 3 set partitions $\{\{1\}, \{2, 3\}\}, \{\{2\}, \{1, 3\}\}, \{\{3\}, \{1, 2\}\} \in \pi_{\{1,\ldots,3\}}$. This number of set partitions in $\pi_{\{1,\ldots,3\}}$ by $\sigma \in \pi_3$ is computed by $\mathrm{sym}((1, 1, 0)) = \frac{n!}{k_1! \cdots k_n! \cdot 1!^{k_1} \cdots n!^{k_n}} = \frac{3!}{1!1!0! \cdot 1!^1 2!^1 3!^0} = 3$. This shows how $\mathrm{sym}(\sigma)$ relates the formula in 17 to the formula in 19

Following the formula in 19 we have exactly the result from 18 with the variables $\partial x_1, \ldots \partial x_n$ identified to $dx^n$:

$$\frac{d^3}{dx^3}f(g(x)) = f^{(1)}(g(x))\frac{d^3 g(x)}{dx^3}$$
$$+ 3 * f^{(2)}(g(x))(\frac{dg(x)}{dx}\frac{d^2 g(x)}{dx^2})$$
$$+ f^{(3)}(g(x))\left(\frac{dg(x)}{dx}\right)^3 \tag{20}$$

Importantly, the Faà di Bruno Formula 19 gives an expression for the $n$th derivative of the composition $f(g(x))$ in terms of (mostly lower order) derivatives of the constituent functions $f$ and $g$. Since we are concerned with computing all derivatives of the composition up to $n$, this formula allows us to share the work of computing lower-order derivatives with all subsequent higher orders.

## C   More on Taylor Polynomials

### C.1   Normalized Taylor Coefficients vs Derivative Coefficients

To clarify the relationship between the presentation in Griewank and Walther [1] and our results we give the distinction between the Taylor coefficients and derivative coefficients, also known, unhelpfully, as *Tensor* coefficients.

For a sufficiently smooth vector valued function $f : \mathbb{R}^n \to \mathbb{R}^m$ and the polynomial

$$x(t) = x_{[0]} + x_{[1]}t + x_{[2]}t^2 + x_{[3]}t^3 + \cdots + x_{[d]}t^d \in \mathbb{R}^n \tag{21}$$

we are interested in the $d$-truncated Taylor expansion

$$y(t) = f(x(t)) + O(t^{d+1}) \tag{22}$$
$$\equiv y_{[0]} + y_{[1]}t + y_{[2]}t^2 + y_{[3]}t^3 + \cdots + y_{[d]}t^d \in \mathbb{R}^m \tag{23}$$

with the notation that $y_{[i]} = \frac{1}{i!}y_i$ is the *Taylor coefficient*, which is the normalized *derivative coefficient* $y_i$.

The Taylor coefficients of the expansion, $y_{[j]}$, are smooth functions of the $i \leq j$ coefficients $x_{[i]}$,

$$y_{[0]} = y_{[0]}(x_{[0]}) \qquad\qquad = f(x_{[0]}) \tag{24}$$
$$y_{[1]} = y_{[1]}(x_{[0]}, x_{[1]}) \qquad\qquad = f'(x_{[0]})x_{[1]} \tag{25}$$
$$y_{[2]} = y_{[2]}(x_{[0]}, x_{[1]}, x_{[2]}) \qquad\qquad = f'(x_{[0]})x_{[2]} + \frac{1}{2}f''(x_{[0]})x_{[1]}x_{[1]} \tag{26}$$
$$y_{[3]} = y_{[3]}(x_{[0]}, x_{[1]}, x_{[2]}, x_{[3]}) \quad = f'(x_{[0]})x_{[3]} + f''(x_{[0]})x_{[1]}x_{[2]} + \frac{1}{6}f'''(x_{[0]})x_{[1]}x_{[1]}x_{[1]} \tag{27}$$
$$\vdots$$

These, as given in Griewank and Walther [1], are written in terms of the normalized, Taylor coefficients. This obscures their direct relationship with the derivatives, which we make explicit.

Consider the polynomial eq. (21) with Taylor coefficients expanded so their normalization is clear. Further, let's use suggestive notation that these coefficients correspond to the higher derivatives of of $x$ with respect to $t$, making $x(t)$ a Taylor polynomial. That is $x_{[i]} = \frac{1}{i!}x_i = \frac{1}{i!}\frac{d^i x}{dt^i}$.

$$x(t) = x_0 + x_1 t + \frac{1}{2!}x_2 t^2 + \frac{1}{3!}x_3 t^3 + \cdots + \frac{1}{d!}x_d t^d \in \mathbb{R}^n \tag{28}$$
$$= x_0 + \frac{dx}{dt}t + \frac{1}{2!}\frac{d^2 x}{dt^2}t^2 + \frac{1}{3!}x_3 t^3 + \cdots + \frac{1}{d!}\frac{d^d x}{dt^d}t^d \in \mathbb{R}^n \tag{29}$$
$$\tag{30}$$

Again, we are interested in the polynomial eq. (23), but with the normalization terms explicit

$$y(t) \equiv y_0 + y_1 t + \frac{1}{2!}y_2 t^2 + \frac{1}{3!}y_3 t^3 + \cdots + \frac{1}{d!}y_d t^d \in \mathbb{R}^m \tag{31}$$

Now we can expand the expressions for the Taylor coefficients $y_{[i]}$ to expressions for derivative coefficients $y_i = i! y_{[i]}$

The coefficients of the Taylor expansion, $y_j$, are smooth functions of the $i \leq j$ coefficients $x_i$,

$$
\begin{aligned}
y_0 = y_0(x_0) \qquad & = y_{[0]}(x_0) \\
& = f(x_0) \qquad\qquad\qquad\qquad\qquad\qquad (32) \\
y_1 = y_1(x_0, x_1) \qquad & = y_{[1]}(x_0, x_1) \\
& = f'(x_0) x_1 \\
& = f'(x_0) \frac{dx}{dt} \qquad\qquad\qquad\qquad\qquad (33) \\
y_2 = y_2(x_0, x_1, x_2) \qquad & = 2! \left( y_{[2]}(x_0, x_1, \frac{1}{2!} x_2) \right) \\
& = 2! \left( f'(x_0) \frac{1}{2!} x_2 + \frac{1}{2} f''(x_0) x_1 x_1 \right) \\
& = f'(x_0) x_2 + f''(x_0) x_1 x_1 \\
& = f'(x_0) \frac{d^2 x}{dt^2} + f''(x_0) \left( \frac{dx}{dt} \right)^2 \qquad\qquad (34) \\
& = \frac{d^2}{dt^2} f(x(t)) \qquad\qquad\qquad\qquad\qquad (35) \\
y_3 = y_3(x_0, x_1, x_2, x_3) \qquad & = 3! \left( y_{[3]}(x_0, x_1, \frac{1}{2!} x_2, \frac{1}{3!} x_3) \right) \\
& = 3! \left( f'(x_0) \frac{1}{3!} x_3 + f''(x_0) x_1 \frac{1}{2!} x_2 + \frac{1}{6} f'''(x_0) x_1 x_1 x_1 \right) \\
& = f'(x_0) x_3 + 3 f''(x_0) x_1 x_2 + f'''(x_0) x_1 x_1 x_1 \\
& = f'(x_0) \frac{d^3 x}{dt^3} + 3 f''(x_0) \frac{dx}{dt} \frac{d^2 x}{dt^2} + f'''(x_0) \left( \frac{dx}{dt} \right)^3 \qquad (36) \\
& = \frac{d^3}{dt^3} f(x(t)) \qquad\qquad\qquad\qquad\qquad (37)
\end{aligned}
$$

$$\vdots$$

Therefore, eqs. (32), (33), (35) and (37) show that the derivative coefficient $y_i$ are exactly the $i$th order higher derivatives of the composition $f(x(t))$ with respect to $t$. The key insight to this exercise is that by writing the derivative coefficients explicitly we reveal that the expressions for the terms, eqs. (32) to (34) and (36), are given by the Faà di Bruno formula eq. (19). For example, notice the equivalence of the expression for $y_3$ in eq. (34) and the example in eq. (20).

# D   Differential Equations

## D.1   Autonomous Form

We can transform the initial value problem

$$\frac{dx}{dt} = f(x(t), t) \quad \text{where} \quad x(t_0) = x_0 \in \mathbb{R}^n \tag{38}$$

into an *autonomous* dynamical system by augmenting the system to include the independent variable with trivial dynamics [7]:

$$\frac{d}{dt}\begin{pmatrix} x \\ t \end{pmatrix} = \begin{pmatrix} f(x(t)) \\ 1 \end{pmatrix} \quad \text{where} \quad \begin{pmatrix} x(0) \\ t(0) \end{pmatrix} = \begin{pmatrix} x_0 \\ t_0 \end{pmatrix} \in \mathbb{R}^n \tag{39}$$

We do this for notational convenience, as well it disambiguates that derivatives with respect to $t$ are meant in the "total" sense. This is aleviates the potential ambiguity of $\frac{\partial}{\partial t} f(x(t), t)$ which could mean both the derivative with respect to the second argument and the derivative through $x(t)$ by the chain rule $\frac{\partial f}{\partial x}\frac{\partial x}{\partial t}$.

## D.2   Recursive ODE Solution with `jet`

Recall that `jet`, by definition (16), gives us the coefficients for $y_i$ as a function of $f$ and the coefficients $x_{j \leq i}$. We can use `jet` and the relationship (8) to recursively compute the coefficients of the solution polynomial.

---

**Algorithm 2** ODE Solution by Recursive Jet

---

```
# Have: x_0, f
# Want: x_1, ..., x_d

y_0 = jet(f, x_0, [0])
x_1 = y_0

for i in range(d):
    (y_0,[y_1,...,y_i]) = jet(f, x0, [x1,..., x_i])
    x_{i+1} = y_i

return x_0, [x_1, ..., x_d]
```

---

## D.3   Coefficient Doubling by Newton's Method

From corollary 13.2 of Griewank and Walther [1] we have that the coefficients depend linearly on the upper half of their input coefficients. Further, the linear dependence is determined by the lower half of the input coefficients. To be clear, coefficient $y_k(x_0, \ldots, x_k)$ depends linearly on its arguments $x_j$ where $j > \frac{k}{2}$. Further, this linear dependency is fully determined by non-linear arguments through Jacobians $A_i$ where $i \leq \frac{k}{2}$.

The dependence is given in Griewank and Walther [1] in terms of Taylor coefficients as for $\frac{k}{2} < j \leq k+1$

$$y_{[k]} = \hat{y}_{[k]}(x_0, \ldots, x_{j-1}, 0, \ldots, 0) + \sum_{i=j}^{k} A_{[k-i]} x_{[i]} \tag{40}$$

Where, for $0 \leq m \leq n$, we have $A_{[m]} = \frac{\partial y_{[n]}}{\partial x_{[n-m]}}$ that are the Jacobians of the $n$th output Taylor coefficient with respect to the $(n-m)$th input Taylor coefficient. Further, the $\hat{y}_{[m]}$ denotes that it is an intermediate quantity which only captures the non-linear dependence on the lower half coefficients, and will be updated with the linear dependence to produce $y_{[m]}$,

To write this in derivative coefficients we expand the factorial terms. In particular,

$$A_{[m]} = \frac{\partial y_{[n]}}{\partial x_{[n-m]}} = \frac{\partial \frac{1}{n!} y_n}{\partial \frac{1}{n-m!} x_{n-m}} = \frac{(n-m)!}{n!} \frac{\partial y_n}{\partial x_{n-m}} = \frac{(n-m)!}{n!} A_m \tag{41}$$

Where $A_m = \frac{\partial y_n}{\partial x_{n-m}}$ are the Jacobians of the $n$th output derivative coefficient with respect to the $(n-m)$th input derivative coefficient. For example

$$\frac{\partial y_0}{\partial x_0} = \frac{\partial y_1}{\partial x_1} = \frac{\partial y_2}{\partial x_2} = A_0 \tag{42}$$

$$\frac{\partial y_1}{\partial x_0} = \frac{\partial y_2}{\partial x_1} = A_1 \tag{43}$$

$$\frac{\partial y_2}{\partial x_0} = A_2 \tag{44}$$

$$\vdots$$

We will later exploit the identification of some Jacobians, e.g. that $\frac{\partial y_0}{\partial x_0} = \frac{\partial y_1}{\partial x_1}$ for performance gains. But for now we will use the result to simplify notation, which is that we will be interested in the particular expression for $A_m$ that is given by the derivative of the $y_{j-1}$ with respect to the $x_m$. That is, we write

$$A_{[m]} = \frac{\partial y_{[j-1]}}{\partial x_{[(j-1)-m]}} = \frac{((j-1)-m)!}{(j-1)!} \frac{\partial y_{j-1}}{\partial x_{(j-1)-m}} = \frac{((j-1)-m)!}{(j-1)!} A_m \tag{45}$$

Now, we can write eq. (40) in terms of derivative coefficients by expanding all factorial terms including the expansion given in eq. (45):

$$\frac{1}{k!} y_k = \frac{1}{k!} \hat{y}_k(x_0, \ldots, x_{j-1}, 0, \ldots, 0) + \sum_{i=j}^{k} A_{[k-i]} \frac{1}{i!} x_i \tag{46}$$

Multiplying all terms by the factorial factor $\frac{1}{k!}$

$$y_k = \hat{y}_k(x_0, \ldots, x_{j-1}, 0, \ldots, 0) + k! \sum_{i=j}^{k} A_{[k-i]} \frac{1}{i!} x_i \tag{47}$$

Using the expansion derived in 45

$$y_k = \hat{y}_k(x_0, \ldots, x_{j-1}, 0, \ldots, 0) + k! \sum_{i=j}^{k} \frac{((j-1)-(k-i))!}{(j-1)!} A_{k-i} \frac{1}{i!} x_i \tag{48}$$

and simplifying

$$y_k = \hat{y}_k(x_0, \ldots, x_{j-1}, 0, \ldots, 0) + \frac{k!}{(j-1)!} \sum_{i=j}^{k} \frac{((j-1)-(k-i))!}{i!} A_{k-i} x_i \tag{49}$$

We make use of two critical properties here:

**Remark 1 (Identification of Coefficient Jacobians)** *By definition $A_j = \frac{\partial y_n}{\partial x_{n-j}}$, this identifies Jacobians of certain output coefficients with respective input coefficients.*

For example, as seen in (42), we can compute $A_0$ either as a derivative of the coefficient $y_0$ with respect to $x_0$ or by the derivative of coefficient $y_1$ with respect to $x_1$.

**Remark 2 (Linear Dependence via Jacobian-vector Products)** *The terms that capture the linear dependence in (46) are the sum of Jacobian-vector products with Jacobians $A_{k-i}$ and vectors $x_i$.*

In particular, note that this means we do not explicitly instantiate the Jacobians $A_{k-i}$ as they are immediately contracted against the vector $x_i$. We can use Forward-mode AD to compute the Jacobian-vector product implicitly.

Griewank and Walther [1] note that we can exploit this linear dependence to more than double the number of coefficients computed for the solution. That is, for $s = 0, 1, \ldots$ we will have $j = 2^{s+1} - 1$ in the expression (49). This will allow us to compute coefficients up to $2j$.

We can make this method clear by example computing the fist 6 coefficients to the ODE solution.

**Example 1 (Computing solution coefficients $x_1, \ldots, x_6$ by Newton Doubling)**

We start with $s = 0$ and $j = 2^{s+1} - 1 = 1$. So we can compute up to coefficient $x_{2j} = x_2$.

We begin by computing the first coefficient $x_1 = y_0$. There is no linear relationship we can exploit here, so this involves computing `jet` at $x_0$ as usual. However, if we also capture the linear dependence, with `jax.linearize`, we can then compute the Jacobian-vector product with newly computed coefficients which will make use of remark 2.

```
f_jet0 = lambda x0 : jet(g,(x0,),((zero_term,),))
(y0,[y1h]), f_jvp = linearize(f_jet0,x0)
x1 = y0      # recurrence relationship
```

In addition to computing y0, the $y_0$ coefficient, we also compute y1h which corresponds to $\hat{y}_1$ in (49).

Now, we use the formula (49) with $k = 1$ and $j = 1$:

$$y_1 = \hat{y}_1(x_0, 0) + \frac{1!}{(1-1)!} \sum_{i=1}^{1} \frac{((1-1) - (1-i))!}{1!} A_{1-i} x_i \qquad (50)$$

$$y_1 = \hat{y}_1(x_0, 0) + A_0 x_1 \qquad (51)$$

We used `jax.linearize` to capture linear dependence of the $y_0$ coefficient with respect to its input, $x_0$. Now `f_jvp(xi)[0]` will compute the Jacobian-vector product $\frac{\partial y_0}{\partial x_0} x_i = A_0 x_i$. This allows us to compute the expression (51).

```
y1 = y1h + f_jvp(x1)
x2 = y1      # recurrence relationship
```

The reccurence relationship gives us $x_2 = y_1$. Now we see with a single call to `jet` we've computed $x1, x2$.

Let's consider computing the next few coefficients to note a further improvement. For this we will have $s = 1$ and $j = 2^{s+1} - 1 = 3$. So we can compute up to the coefficient $x_{2j} = x_6$

eq. (46) tells us that coefficients up to $x_5$ will depend non-linearly on $x_0, x_1, x_2$ plus linear updates involving $A_0$ and $A_1$

```
f_jet012 = lambda x0,x1,: jet(g, (x0,), ([x1,x2] + [zero_term]*3,))
(y0,[y1, y2, y3h, y4h, y5h]), f_jvp = linearize(f_jet01, x0, x1)
x3 = y2      # recurrence relationship
```

Note that at this stage the function which computes the Jacobian-vector products, `f_jvp`, now takes 2 arguments. Where `[1][0]` corresponds to indexing the $y_1$ coefficient, if we call `f_jvp(xi, zero_term)[1][0]`, we compute $\frac{\partial y_1}{\partial x_0} x_i = A_1 x_i$. We can now make use of remark 1 to notice that $A_0 x_i = \frac{\partial y_0}{\partial x_0} x_i = \frac{\partial y_1}{\partial x_1} x_i$. We could compute this by indexing into the $y_0$ coefficient, and calling `f_jvp(xi, zero_term)[0]`. However, we can equivalently compute this by remaining indexed into the $y_1$ coefficient and computing the derivative with respect to the other input `f_jvp(zero_term, xi)[1][0]`.

```
y3 = y3h + f_jvp(zero_term, x3)[1][0]
x4 = y3      # recurrence relationship
```

The use of remark 1 in the above coefficient is inconsequential. However, in computing the coefficient $y_4$ it will be offer performance improvement.

## D.4 Linearity of Jacobian-Vector Products

In particular, notice that linear updates are given by a sum of Jacobian-vector products $\sum_{i=j}^{k} A_{k-i} x_i$. Note that all $A_{k-i}$ are a Jacobian of $y_{j-1}$ with respect to different inputs $x_{(j-1)-(k-i)}$. So all Jacobians appearing in the sum are derivatives with respect to a different input of a multivariate function $y_{j-1}(x_0, \ldots, x_{j-1})$. We can use this fact, together with the linearity of Jacobian-vector products, to compute the sum in one single Jacobian-vector product, instead of the sum of many Jacobian-vector products.

This allows us to exploit the linearity of Jacobian-vector products, specifically, consider the Jacobian of the multivariate input function $f([x_1, x_2])$

Specifically, the linear property we're exploiting can be seen by considering Jacobian-vector products of a multivariate function $f([x_1, x_2])$. The sum of Jacobian-vector products for different vectors is equal to the Jacobian-vector product on the sum of the vectors. In particular, if those vectors only have one non-zero element in the $i$th index, then they correspond to derivatives with respect to the $i$th argument of $f$:

$$\frac{\partial f([x_1, x_2])}{\partial [x_1, x_2]} \begin{bmatrix} x_i \\ 0 \end{bmatrix} + \frac{\partial f([x_1, x_2])}{\partial [x_1, x_2]} \begin{bmatrix} 0 \\ x_j \end{bmatrix} = \frac{\partial f([x_1, x_2])}{\partial [x_1, x_2]} \begin{bmatrix} x_i \\ x_j \end{bmatrix} \tag{52}$$

In particular, we can exploit this fact by taking a single Jacobian-vector product with respect to all the input coefficients at once, and the resulting value will be the sum of their individual Jacobian-vector products, as desired.

