# OpenReview forum: "Taylor-Mode Automatic Differentiation for Higher-Order Derivatives in JAX"
_NeurIPS.cc/2019/Workshop/Program_Transformations — Program Transformations @NeurIPS2019 Oral_

### Official Review · AnonReviewer1 · 2019-09-29
**Higher-order derivatives without nested automatic differentiation**

**Confidence:** 4
**Rating:** 8

**Review:**

The authors present their implementation of Taylor-mode automatic differentiation (AD) in the JAX library. Taylor-mode refers to the computation of derivatives using the higher-order chain rule (also known as Faà di Bruno's formula) in a forward AD setting where higher-order derivatives up to a chosen level are propagated simultaneously during the forward evaluation of a function. This yields higher-order derivatives with better scaling compared with the alternative approach of nested application of first-order AD operations.

One of the advantages of this mode is the ability to exploit shared evaluation, meaning that terms computed for lower-level derivatives might also show up in the higher-levels and therefore do not need to be computed more than once as all levels are computed in the same shared evaluation. This is nicely explained in the paper, but I’m also curious whether the same outcome (shared evaluation) could also be achieved in a source-transformation-based nested AD case combined with compiler optimizations noticing and exploiting shared terms in the resulting higher-order derivative code.

I think this paper clearly belongs in this workshop. The technique implemented has been known in the automatic differentiation literature (e.g., Griewank and Walther, 2008) as the authors clearly mention, but not so much by the machine learning audience of this workshop.

I have the following (mostly minor) comments to improve the paper in the camera-ready version:

- I think a brief explanation of “what is JAX?” is needed to aid the general reader, and you need to provide a reference or at least a URL of the project.
- In the notation of, e.g., “\partialf(x)[v]”, it would be better if you introduced what exactly the square brackets are used for.
- I think the symbol “K” in equations 1 and 2 corresponds to “k” elsewhere in the paper. Also in equations 4 and 5, the “end” in “\sigma_end” also refers to “k” if I’m not mistaken. It would be good to use the same symbol everywhere.
- In Section 2, starting the introduction with the case of “f = g o h” and then continuing the rest of the section and appendices A and B with the case of “f o g” is slightly confusing and not ideal. I think the beginning of Section 2 can be improved in general.
- In Section 4 you say “dynamical systems are defined directly in terms of their derivatives”. I think this is correct for flows, but it is an unnecessary generalization. There are dynamical systems that are not defined in terms of derivatives, such as maps, discrete-time systems, etc.
- The paper ends abruptly after Section 4. It would really help to have some unifying conclusion.

---

### Official Review · AnonReviewer2 · 2019-09-30
**Sweats the details of multivariate higher-order forward mode**

**Confidence:** 4
**Rating:** 8

**Review:**

“The naïve approach to higher order differentiation, supported by most AD frameworks, is to compose derivative operation until the desired order is achieved.”  I think that’s awfully optimistic: my impression is that most AD system either don’t handle nesting at all, or manifest all sorts of bugs (both numeric incorrectness and surprising inefficiency) when nesting is used.

I think, but am not absolutely positive, that some of the equations and definitions in Pearlmutter and Siskind (POPL-2007, doi:10.1145/1190215.1190242), if truncated at 𝒪(𝜀ⁿ), are equivalent to some of the formulas used here. On the other hand, they leave some symmetries on the floor.  Basically, they assume different nested derivatives are always on unique variables, whereas that situation is focused on here and the redundancies caused by permutational symmetries are exploited. It does seem like even when distinct variables are in play, some clever factorization might make the symmetries manifest and allow them to be exploited anyway.

Some of the formulas here are in "big sum" form when they could instead be formulated recursively while giving identical (or sometimes superior, due to factoring out comment subexpressions) numeric operation counts. The big sum is traditional in calculus texts. But, the recursive formulation lends itself to more concise implementation.

- ref [6], title contains proper name which should be capitalized, and maybe a missing accent

Recommendation:

This is clearly on-topic. Might be a bit technical for an oral. In any case, an oral (or poster, for that matter) would beg for a bunch of color-coded examples to really show the way things get redundant in a naïve formulation, and how this can be avoided by the right book-keeping.

---

### Public Comment · ~Andreas_Griewank2 · 2019-10-02
**Some basic observations missing**

I agree with one of the other reviewers that generally applying AD tools (whether source transformation or operator overloading) in a nested fashion will either not work at all or be frightfully inefficient.
Of course nesting forward differentiation leads mathematically to the Faa di Bruno formula, which involves many small loops.
The more interesing question is what happens if one composes the reverse mode with itself or with the forward mode. For example the loss function may already contain a (usually moderate) number of derivatives of the predictions function, which can be computed in the forward mode, and then one wants to differentiate the empirical risk with respect to a rather large number of weights and other model parameters.  It is well known GriewankWalther that and how one can always get by with one reverse sweep and the associate storage of the forward trajectory.
The authors seem to assume that one always wants a tetrahedral tensor, i.e. all partial derivatives up to a certain total order k. In the situation sketched above that is not the case.  Moreover, it is surprising that the authors do no consider the possiblility of replacing the computation of one  multivariate Taylor polynomial with the propagation of a family of univariate Taylor polynomials (see e.g. GriewankUtkeWalther)  from whose coefficients  the whole derivative tensor can be reconstructed at a negligible effort.  If fast convolution methods are used  the total effort can be limited to O{dˆˆ^(k-1) (log k) /(k-1)!} arithmetic operations , always times the effort for evaluating the . As far as I know this possibility awaits an efficient implementation. For multivariate forward propagation the complexity growth can be estimated by  that of multiplyng to tensors of dimension d of order k, namely  [(2k+d) choose d] approx  (2d)ˆk /k!. Please some statements on complexity.
As an AD oldie I am often inclined to mutter "We have done all that". However, except for the beam physics applications of Martin Berz etal there have not been that many applications of higher order derivatives.  So I would be delighted if it happens in AI. There are plenty of more references in the Special Issue of OMS on Advances in Algorithmic Differentiation.

---

### Public Comment · ~Jeffrey_Mark_Siskind1 · 2019-10-02
**lack of novelty**

FADBAD++ had TADIFF which implemented a forward mode Taylor tower.

http://www.autodiff.org/?module=Tools&tool=FADBAD%2FTADIFF
http://www.autodiff.org/?module=Publications&submenu=list%20publications&id=bendtsen1996faf
https://pdfs.semanticscholar.org/ad56/94d44e1f065d06e41a34a62e86f5f1d25ffb.pdf?_ga=2.182337959.1602511013.1570049134-1142852153.1541019591
https://pdfs.semanticscholar.org/d84a/38d06e121e216a0600ad7cd6a9617001d7a1.pdf?_ga=2.119384457.1602511013.1570049134-1142852153.1541019591

Karczmarczuk (1998, 2001) presented a lazy forward mode Taylor tower.

http://www.autodiff.org/?module=Publications&submenu=list%20publications&id=karczmarczuk1998fdo
http://www.autodiff.org/?module=Publications&submenu=list%20publications&id=karczmarczuk2001fdo
https://dl.acm.org/citation.cfm?id=289442
https://link.springer.com/article/10.1023/A:1011501232197

Pearlmutter & Siskind (2007) presented a lazy forward mode Taylor tower that
nested and was called via a higher-order function.

http://www.autodiff.org/?module=Publications&submenu=list%20publications&id=pearlmutter2007lmh

Mu Wang's PhD thesis with Alex Pothen presented higher-order reverse mode that
handled sparsity and symmetry. Preliminary versions supporting second-order
(Hesssions) were published. This work was even published in this very venue
two year ago.

http://www.autodiff.org/?module=Publications&submenu=list%20publications&id=wang2016col
http://www.autodiff.org/?module=Publications&submenu=list%20publications&id=wang2016epi
https://openreview.net/forum?id=Hkmj6tzRZ
https://docs.lib.purdue.edu/dissertations/AAI10270057/

Nothing in this paper is novel. It is embarrassing that it was accepted.

---

> ### Public Comment · ~Matt_Johnson1 · 2019-10-04
> **Thanks for the refs and blunt feedback, not claiming novelty**
>
> Thanks, Jeff, for the references and blunt feedback!
>
> We don’t claim novelty: that’s why we wrote in the abstract “This paper describes a more efficient method, already known but with a new presentation, and its implementation in JAX.” Maybe we can improve the text to emphasize the “already known” part more. Suggestions welcome!
>
> In any case it’s very helpful to get a thorough list of pointers, and we can update the text to cite them.
>
> We think this is a good technique to discuss in the ML community now, especially in the context of neural ODEs, which is why we wanted to bring it up at this workshop. And learning more connections to the autodiff literature is one of the good outcomes we wanted. So thank you for that! My understanding is that this is exactly what NeurIPS workshops are for.
>
> In any case, I’m interested to discuss more, either on this thread or (even better) in person at the workshop if you’ll be there. I think there are some pieces here, not unpacked in this short paper, that you’ll find quite interesting, or at the very least not embarrassing :) I hope to win you over soon!

---

### Decision · Program_Chairs · 2019-10-01

**Decision:**

Accept (Oral)

**Comment:**

This paper contributes strongly to bridging the gaps between the AD and ML communities, which is a primary goal of this workshop.